# Structural insight into the mechanism of energy transfer in cyanobacterial phycobilisomes

Lvqin Zheng[1,5], Zhenggao Zheng[2,3,5], Xiying Li[2,5], Guopeng Wang[1], Kun Zhang [2], Peijun Wei[2], Jindong Zhao [2,4✉] & Ning Gao [1✉]

Phycobilisomes (PBS) are the major light-harvesting machineries for photosynthesis in cyanobacteria and red algae and they have a hierarchical structure of a core and peripheral rods, with both consisting of phycobiliproteins and linker proteins. Here we report the cryo-EM structures of PBS from two cyanobacterial species, *Anabaena* 7120 and *Synechococcus* 7002. Both PBS are hemidiscoidal in shape and share a common triangular core structure. While the *Anabaena* PBS has two additional hexamers in the core linked by the 4th linker domain of ApcE ($L_{CM}$). The PBS structures predict that, compared with the PBS from red algae, the cyanobacterial PBS could have more direct routes for energy transfer to ApcD. Structure-based systematic mutagenesis analysis of the chromophore environment of ApcD and ApcF subunits reveals that aromatic residues are critical to excitation energy transfer (EET). The structures also suggest that the linker protein could actively participate in the process of EET in both rods and the cores. These results provide insights into the organization of chromophores and the mechanisms of EET within cyanobacterial PBS.

[1] State Key Laboratory of Membrane Biology, National Biomedical Imaging Center, Peking-Tsinghua Center for Life Sciences, School of Life Sciences, Peking University, 100871 Beijing, China. [2] State Key Laboratory of Protein and Plant Genetic Engineering, School of Life Sciences, Peking University, 100871 Beijing, China. [3] College of Life Science, Qingdao University, 266071 Qingdao, China. [4] Key Laboratory of Phycology of CAS, Institute of Hydrobiology, Chinese Academy of Sciences, 430072 Wuhan, Hubei, China. [5]These authors contributed equally: Lvqin Zheng, Zhenggao Zheng, Xiying Li. ✉email: jzhao@pku.edu.cn; gaon@pku.edu.cn

The cyanobacteria were responsible for the rise of oxygen on earth about 2.4 billion years ago and they are one of the most important groups of organisms in carbon and nitrogen cycles in the biosphere[1–3]. The major light-harvesting system for solar energy capture in cyanobacteria and red algae is phycobilisome (PBS)[4–8], which consists of phycobiliproteins (PBP) with covalently attached open-chain tetrapyrroles as chromophores (bilins) and linker proteins[5,9,10]. The basic unit of the PBS is a heterodimer of an α subunit and a β subunit (often called αβ monomer). Three αβ monomers form a ring-shaped trimer, and a hexamer is formed by a face-to-face stacking of two trimers. The hexamers are organized further into a highly ordered supramolecular complex of PBS with a nearly unity excitation energy transfer (EET) efficiency.

Two recently determined cryo-EM structures of PBS from red algae[11,12] revealed how the linker proteins organize the PBP into an ordered hierarchical architecture. In general, PBS found in nature is composed of two parts, a central core and peripheral rods that are attached to the core. With a few exception[13], most cyanobacterial PBS have a hemidiscoidal shape and their peripheral rods are variable in length[5]. At present, the cyanobacterial PBS structure of high resolution is unavailable. To gain insights into the mechanism of EET in cyanobacterial PBS, we determined the cryo-EM structures of two cyanobacterial PBS, one with a tri-cylindrical core from *Synechococcus* sp. PCC 7002 (*Synechococcus* 7002) and the other with a penta-cylindrical core[14,15] from *Anabaena* sp. PCC 7120 (*Anabaena* 7120).

## Results

**Overall structures of the cyanobacterial PBS.** PBS complexes were purified from the cultures of *Synechococcus* 7002 and *Anabaena* 7120 following standard procedures[15], and the samples were analyzed biochemically and spectroscopically for their compositional and functional integrity (Supplementary Fig. 1). Both PBS contain phycocyanin (PC), allophycocyanin (APC), and linker proteins but lack phycoerythrin in red algal PBS. Negative-staining electron microscopy (nsEM) analysis revealed a large variation in the length of peripheral rods (Supplementary Fig. 2a, b), and the longest rods could contain as many as six PC hexamers. Initial cryo-EM visualization indicated that the rods are unstable and the distal rod hexamers could readily dissociate from the PBS during cryo-grid preparation. To minimize the PBS complex disassembly, the samples were mildly cross-linked with a low concentration of glutaraldehyde (0.0125% and 0.005% for the *Synechococcus* 7002 and *Anabaena* 7120 PBS samples, respectively) before cryo-freezing. Because the purification buffer contained a high concentration of salt (0.75 M K + /Na + ) required for PBS stabilization, which produced undesired high background noise during cryo-EM imaging, an instant on-grid dilution of samples with a salt-free buffer was applied during grid preparation. With these optimizations of sample preparation, we determined the overall structures of PBS from *Synechococcus* 7002 and *Anabaena* 7120 at resolutions of 3.5 and 3.9 Å, respectively (Supplementary Figs. 3 and 4). As shown in Fig. 1, both PBS are hemidiscoidal in a twofold symmetry (Fig. 1a, e) with the cores located at the center and the peripheral rods attached to the cores. In addition to the rod length variation, as shown in typical 2D average images of side views (Supplementary Figs. 3c and 4c), the relative orientation of the rods to the core is also variable and the densities of the distal rod trimers become highly fragmented in the final cryo-EM maps. Therefore, we conducted our analysis of cyanobacterial PBS structures with rod length within two hexamers (Fig. 1a, b, e, f).

For *Synechococcus* 7002 PBS, we modeled 36 APC αβ monomers in the core, 72 PC αβ monomers in the rods, 6 rod linker proteins (L$_R$), 6 rod-core linker proteins (L$_{RC}$), and 6 core linker proteins (ApcC, formerly L$_C$) in the map. The dimensions of the PBS from *Synechococcus* 7002 are ~450 Å in length, ~ 300 Å in height, and ~220 Å in thickness (Fig. 1a, b). There are six peripheral rods (rods R1/R1', R2/R2', and R3/R3') and each rod contains two hexamers in our model (Fig. 1a, b). The central core contains three cylinders arranged as an equilateral triangle: cylinders A and A' at the bottom and cylinder B on the top (Fig. 1a, b). All three cylinders are each composed of two hexamers (four trimers) joined together in a back-to-back fashion.

For *Anabaena* 7120 PBS, we modeled 48 APC αβ monomers in the core, 84 PC αβ monomers in the rods, 6 rod linker proteins (L$_R$), 8 rod-core linker proteins (three different L$_{RC}$), and 8 core linker proteins (ApcC). The dimensions of *Anabaena* 7120 PBS are ~540 Å in length, ~320 Å in height, and ~210 Å in thickness (Fig. 1e, f). The basic triangular core (Fig. 1a) observed in *Synechococcus* 7002 is also present in *Anabaena* 7120. But *Anabaena* 7120 PBS core has two additional half-cylinders (C and C', one APC hexamer for each), sitting perpendicularly on the shoulders of the two basal core cylinders A and A', respectively (Fig. 1e). Among the eight modeled peripheral rods, six of them (Rb/Rb', Rs1/Rs1', and Rt/Rt') consist of two phycocyanin (PC) hexamers and two (Rs2/Rs2') consist of one PC hexamer.

**The core structures.** The central triangular core consists of two basal cylinders (A/A') that are arranged antiparallelly and a top cylinder B (Fig. 1). The APC αβ monomers in the core cylinders consist of ApcA, ApcB, ApcD, ApcF, and the α$^{LCM}$ of ApcE (Fig. 1 and Supplementary Fig. 2c, d). ApcD and α$^{LCM}$ of ApcE are variants of ApcA, while ApcF is a variant of ApcB[16–18]. ApcE and ApcD are key terminal emitters in EET to photosystem II (PSII) and photosystem I (PSI), respectively[5,19,20], and the excitation energy distribution between PSI and PSII is regulated by a mechanism called state transitions[21,22].

Each core cylinder contains four layers of αβ trimers[15], and ApcF and α$^{LCM}$ of ApcE are located on the 3rd trimer of the basal cylinders (A/A') while ApcD is located on the 4th trimer of the basal cylinders (Supplementary Fig. 2c, d), based on the previous nomenclature[5,15]. In both basal cylinders of the cores, the third layers that contain α$^{LCM}$ of ApcE are protruding toward the thylakoid membrane (Figs. 1a, e and 2c, e) and they are likely critical to the interaction between PBS and PSII[15,20,23]. Notably, the structure of red algal PBS contains three layers of αβ trimers in the basal cylinders and two layers of αβ trimers in the top cylinders[11,12] (Supplementary Fig. 2e). The structural comparison indicates that the equivalent layers of A1/A'1 in the basal cylinders and layers of B1/B1' in the top cylinder of the cyanobacterial PBS are lost in the red algal PBS (Supplementary Fig. 2c–e), likely reflecting an adaptation to the environment of deep ocean water where the more blue–green light is available[11,12]. Despite these differences, the general spatial distributions of ApcE, ApcD and ApcF in the core are the same in the cyanobacteria and red algae PBS. Considering the highly conserved function of PBS to transfer energy to both PSI and PSII, it is not surprising that the basic structure of the PBS core remains unchanged in evolution from cyanobacteria to the chloroplasts of red algae.

We did not observe any orange carotenoid protein (OCP), which provides photoprotection for the cyanobacteria[24–26] in the cores of either PBS. This is an expected result since the cells we used for PBS isolation were not under any light stress. The determination of the core structures of PBS from these cyanobacteria, especially the one from high light tolerant *Synechococcus* 7002, will help future studies of the mechanism of OCPs.

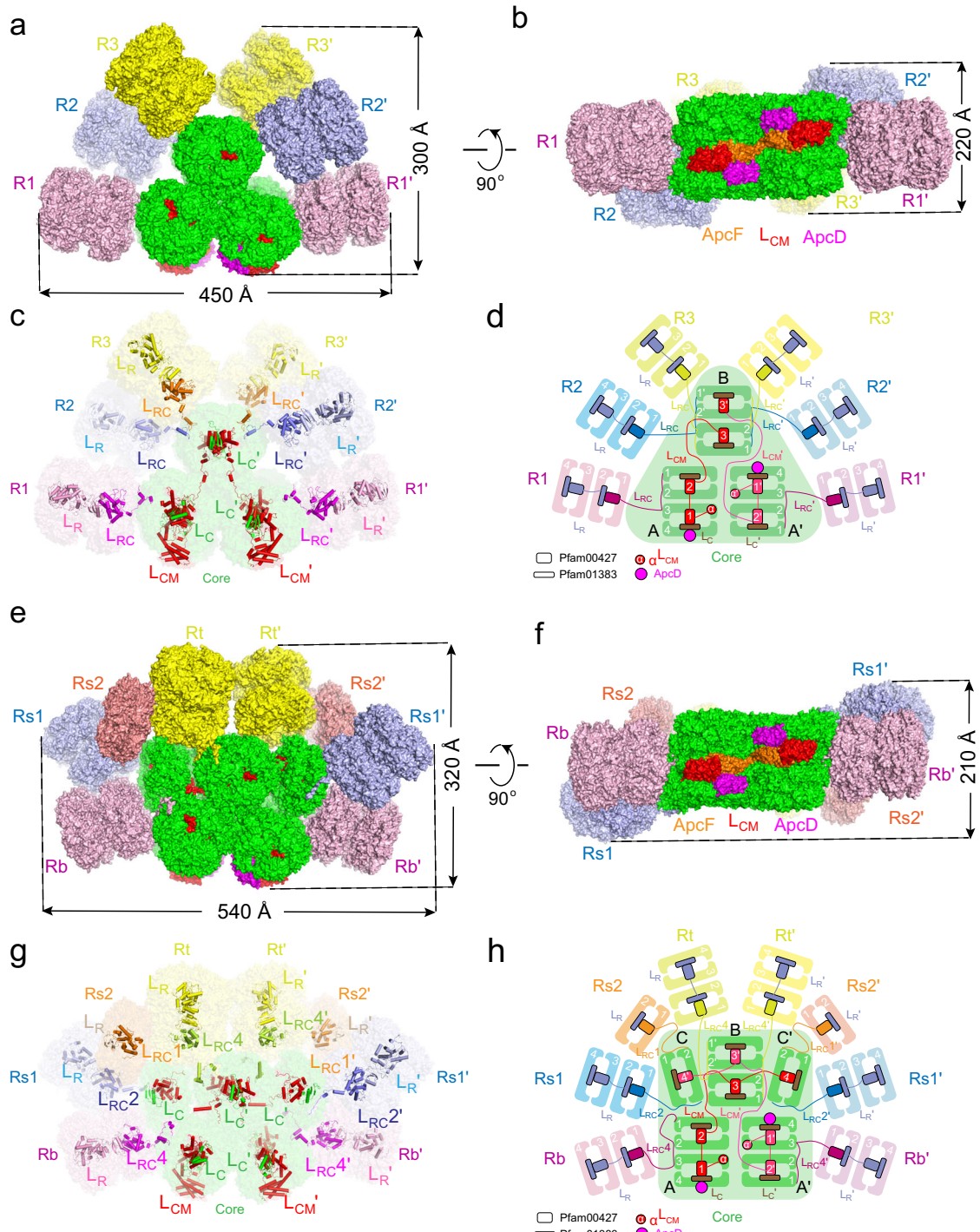

**Fig. 1 Overall structure of the PBS from *Synechococcus* 7002 and *Anabaena* 7120. a** Side view of the PBS from *Synechococcus* 7002. The cryo-EM map is displayed in surface representation, with the core and peripheral rods separately colored. Allophycocyanin trimer, green; phycocyanin, pink, sky blue, and yellow. **b** Bottom view of the PBS from *Synechococcus* 7002. ApcF, ApcD, and $L_{CM}$ are highlighted in orange, magenta, and red, respectively. **c** Distribution of linker proteins in the PBS from *Synechococcus* 7002. Models of linker proteins are shown in cartoon representation and separately colored. **d** Schematic models of the *Synechococcus* 7002 PBS architecture. The connections between PBS components are shown. APC trimer, green; PC trimer, pink, sky blue and yellow. The Pfam00427 and Pfam01383 domains of linker proteins are schematically represented as large and small rectangular boxes, respectively. **e–h** Same as **a–d**, but for the PBS from *Anabaena* 7120.

**Linker proteins**. Four types of linker proteins, rod linker protein ($L_R$), rod-core linker protein ($L_{RC}$), core linker protein (ApcC), and the core-membrane linker protein ($L_{CM}$), are present in the cyanobacterial PBS and they are responsible for organizing PBP into a highly ordered architectural PBS. Compared to PBS of red algae, there is only one $L_R$ protein in the cyanobacterial PBS, and

it shares a generally similar structure with the $L_R1$ of red algae[11,12]. An important difference between the cyanobacterial and red algal $L_R$ proteins is that the cyanobacterial $L_R$ is more compact because the two loops located in the Pfam01383 domain are shorter (Fig. 2a and Supplementary Figs. 5 and 6a, h). We also observed Pfam01383 domains at the second hexamers of each rod

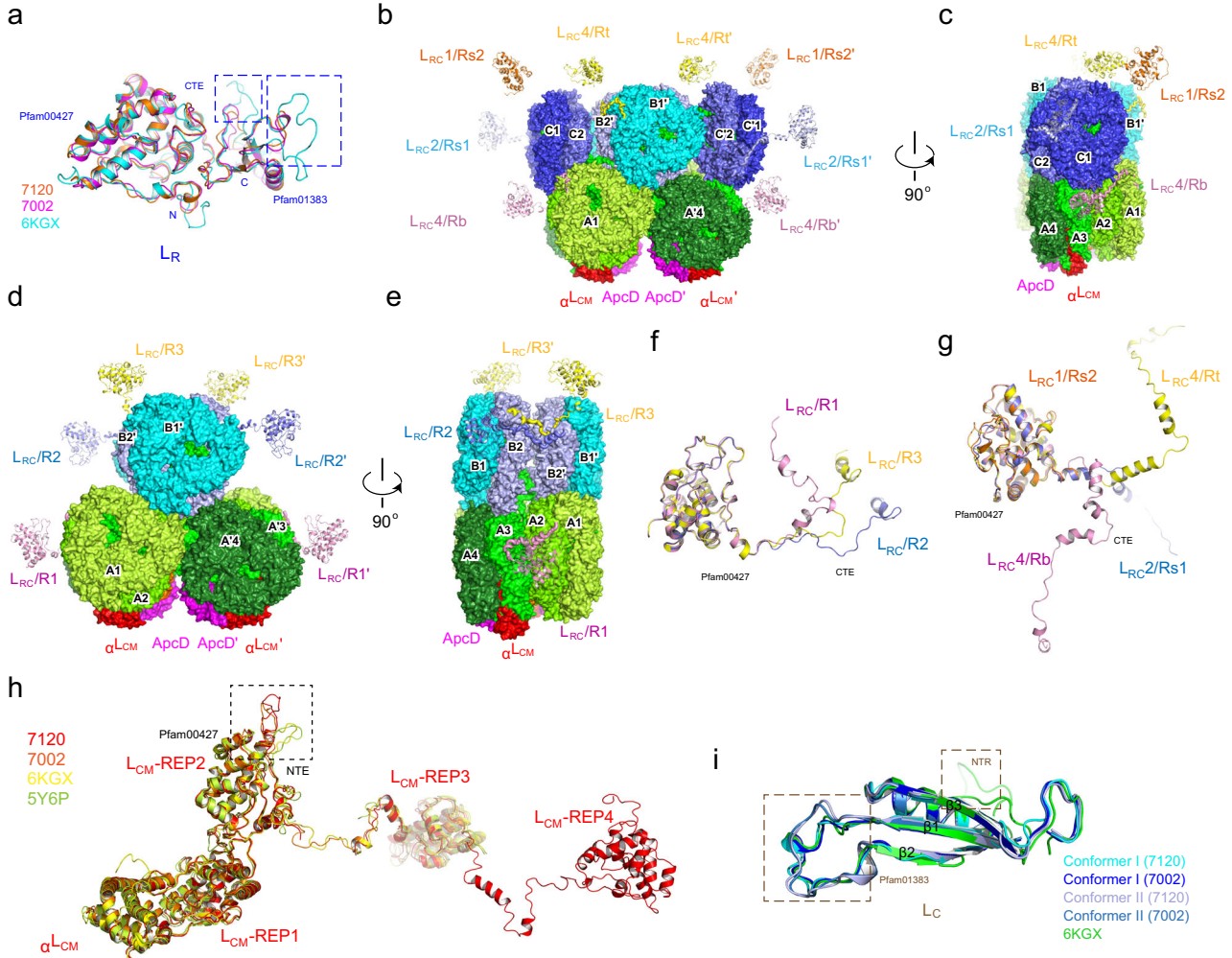

**Fig. 2 Structural comparison of linker proteins. a** Structural comparison of $L_R$ proteins from *Synechococcus* 7002, *Anabaena* 7120 and red algae (PDB 6KGX). The regions with a large difference are highlighted in blue boxes. **b** Spatial distribution of $L_{RC}$ proteins relative to the PBS core of *Anabaena* 7120. The $L_{RC}$ proteins and the core cylinders are shown in cartoon and surface representation, respectively. **c** A 90°-rotated view compared to (**b**). **d** Spatial distribution of $L_{RC}$ proteins relative to the PBS core of *Synechococcus* 7002. **e** A 90°-rotated view compared to (**d**). **f, g** Structural comparison of $L_{RC}$ proteins from *Synechococcus* 7002 (**f**) and *Anabaena* 7120 (**g**). **h** Structural comparison of $L_{CM}$ proteins of *Synechococcus* 7002, *Anabaena* 7120, and red algae (PDB 6KGX and 5Y6P). **i** Structural comparison of $L_C$ proteins of *Synechococcus* 7002, *Anabaena* 7120, and red algae (PDB 6KGX). Two different conformations of $L_C$ are present in our models of PBS from *Synechococcus* 7002 and *Anabaena* 7120.

and confirmed that they all belong to $L_R$ proteins, not from CpcD proteins or ferredoxin: NADP$^+$ reductase (FNR, an enzyme that transfers electrons from ferredoxin/flavodoxin to NADP$^+$ to generate NADPH) both of which also possess a Pfam01383 domain[27–29]. This result was expected since we only modeled two hexamers in each rod while FNR is likely located at the termini of the peripheral rods.

In an attempt to observe FNR on PBS, PBS was isolated from a mutant of *Synechococcus* 7002 lacking the *cpcC* gene encoding the rod linker $L_R$ (strain ΔLr). The peripheral rods of the ΔLr-PBS contain only one hexamer[28,30] (Supplementary Fig. 7) and were expected to bind FNR. However, no FNR was observed in the density map of the ΔLr-PBS (Supplementary Fig. 7). Examination of the density map revealed that, unlike the rods of the WT PBS, no Pfam01383 domain of any source was observed in the cavities of the rod hexamers from the ΔLr-PBS (Supplementary Fig. 7g). Because FNR was identified in the PBS preparations (Supplementary Fig. 1f), the above results suggest that FNR could have dissociated from the rods during cryo-EM sample preparation. A possibility cannot be excluded that the protein FNR on ΔLr-PBS wobbles so much that it could not be observed by cryo-EM.

The rod-core linker proteins, $L_{RC}$, are responsible for the attachment of the peripheral rods to the core. In general, $L_{RC}$ proteins contain a conserved N-terminal Pfam00427 domain, which is located within the cavity of the core-proximal hexamers of the rods, and a C-terminal extension (CTE), which is responsible for interaction with the core[11,12]. For *Synechococcus* 7002 PBS, there is only one type of $L_{RC}$ protein and each PBS has six copies to attach the six rods to the core. These $L_{RC}$ proteins have three different structural conformations in their CTEs to adopt three local core structures for rod attachment (Fig. 2f and Supplementary Fig. 6b–d). The CTE of $L_{RC}$ in R1 (conformer I) interacts with the basal cylinder A on the surface of the trimer layers A3, A2, and A1 (Figs. 1d and 2d, e). In contrast to the basal A/A' cylinders, the top cylinder B interacts with four $L_{RC}$ proteins of two conformations (conformers II and III). On one side of cylinder B, the CTEs of $L_{RC}$/R2 and $L_{RC}$/R3 run in the opposite direction on the surface of B2 and B2' layers (Fig. 2d, e).

The presence of cylinders C/C' in the core of the *Anabaena* 7120 PBS provides additional sites for attachment of more peripheral rods and three types of $L_{RC}$ (two $L_{RC}$1, two $L_{RC}$2, and four $L_{RC}$4) proteins are needed for attaching eight rods to the

core[15]. The rods Rb/Rb' and Rt/Rt' attach to the core through four $L_{RC}4$ proteins in two distinct conformations (Fig. 2b, c, g and Supplementary Fig. 6i–l). For Rb/Rb', the CTE of $L_{RC}4$ stretches along the surfaces of the layers A3/A'3 and A2/A'2 and eventually arrives at the layer A1/A'1 after sliding into the cleft formed by the cylinders A/A' and cylinders C/C' (Fig. 2b, c). In contrast, the CTE of $L_{RC}4$ in Rt/Rt' makes extensive contacts with the cylinders B/B' from the layers of B2/B'2 to B1/B'1 (Fig. 2b). The cylinder C/C' provides a firm platform for attachment of the rods Rs1/Rs1' and Rs2/Rs2' to the core (Fig. 2b, c). Interestingly, no individual hexamers of Rs2/Rs2' present physical contact with cylinders C/C' and the connections are only established via the CTEs of linker $L_{RC}1$, which interact with both the first layer of Rt1/Rt'1 and the cylinders C/C' (Fig. 1h). For Rs1/Rs1', the linker $L_{RC}2$ displays extensive interactions directly with the cylinders C/C' (Fig. 2b, c). In general, more regions of the CTE of the $L_{RC}$ proteins in cyanobacterial PBS are involved in interactions with the core than that of red algal $L_{RC}$ proteins[11,12].

The overall structure of the PBS core is organized by the core-membrane linker $L_{CM}$ in both cyanobacterial and red algal PBS[5,11,12,16,31]. Functionally, $L_{CM}$ is key to the energy transfer from PBS to PSII. Structurally, $L_{CM}$ has a N-terminal $\alpha^{LCM}$ domain (the α domain in $L_{CM}$) followed by repetitive (REP) domains[5]. $L_{CM}$ in the PBS of *Synechococcus* 7002 contains three REP domains, similar to the $L_{CM}$ protein in red algae[11,12]. An additional REP domain (REP4) in its $L_{CM}$ protein is required for the cylinder C/C' in the PBS core of *Anabaena* 7120. (Fig. 2h and Supplementary Fig. 6e, m). One structural difference between the cyanobacterial and red algal $L_{CM}$ proteins is the conformation of a loop located NTE of REP2 (Fig. 2h). This loop in cyanobacterial $L_{CM}$ is critical to the binding of the trimers A1/A'1, which are absent in the red algal PBS core.

The core linker ApcC acts as a plug in the central holes formed by the APC trimers at both ends of core cylinders and plays a role of protection for the core cylinders[32]. While a single Pfam01383 domain is present in both the cyanobacterial and red algal ApcC proteins (Supplementary Fig. 8), cyanobacterial ApcC lacks N-terminal extension possessed by red algal ApcC (Fig. 2i and Supplementary Fig. 8). The three core cylinders of both cyanobacterial PBS contain four trimers and each cylinder possesses two ApcC proteins at layers of A1/A'1, A4/A'4, and B1/B1'. The two shoulder half-cylinders in *Anabaena* 7120 only contain one ApcC each at layers of C1/C'1. In contrast, in the red algal PBS each three-layered basal core cylinder has only one ApcC in the equivalent layer of A4/A'4 of cyanobacterial PBS[11,12]. Structurally, the red algal ApcC shares a similar conformation in the Pfam01383 domain with ApcC proteins from the equivalent positions in cyanobacterial PBS (Fig. 2i, Conformer I and Supplementary Fig. 6f, n). The four additional ApcC linkers in cyanobacterial triangular cores (located at the trimers of B1, B'1, A1, and A'1) and the two extra ApcC proteins in half-cylinders (C1 and C'1) of *Anabaena* 7120 share another conformation, different from the ones located at A4 and A'4 (Fig. 2i, Conformer II and Supplementary Fig. 6g, o).

**Bilins distribution in PBS core**. Bilins are open-chain tetra-pyrrole molecules covalently attached to PBP and they are responsible for the light absorption of PBS. Among four types of bilins found in PBS, only phycocyanobilin (PCB) is present in the PBS of this study. We defined a total of 288 PCB in *Synechococcus* 7002 PBS with 72 PCB in the core and 216 PCB in the rods (Fig. 3a). In *Anabaena* 7120, we defined a total of 348 PCB, with 96 PCB in the core, and 252 PCB in the rods (Fig. 3d). The overall spatial arrangement of the bilins in the cores is conserved from the cyanobacterial hemidiscoidal to the ellipsoidal and the block-type PBS in red algae[11,12]. Excitation received from rods or within the core migrates through bilins in the core and eventually arrives at one of the terminal emitters, $\alpha^{LCM}$ and ApcD, which transfer the energy to reaction centers of PSII and PSI. Because distances between the chromophores decide the EET efficiency[33], the most probable paths of EET to the terminal emitters are deduced based on the shortest distances among the bilin pairs (Fig. 3 and Supplementary Fig. 9). While the routes for EET to $\alpha^{LCM}$ within the cores that go through the bilin of ApcF are conserved in both the cyanobacterial and red algal PBS cores, the EET to the other terminal emitter ApcD is different between the cyanobacterial and red algal PBS cores. Besides the cross-trimer routes for EET to ApcD[11] present in both cyanobacterial and red algal PBS cores, a more direct energy transfer to ApcD exists in the cyanobacterial PBS because of the presence of the additional trimers B1/B1' in cylinder B of the cores of the cyanobacterial PBS, compared to the red algal PBS (Fig. 3 and Supplementary Fig. 10). In *Synechococcus* 7002 PBS, energy from R2 and R3 could be transferred to ApcD on A4 and A'4 through trimers of B1 and B1' in cylinder B, respectively (Fig. 3b, c). In *Anabaena* 7120 PBS, the presence of half-cylinders C/C' in the core pushes rods Rb/Rb' toward trimers A4/A'4, and energy from Rb could be directly transferred to ApcD in trimer A4 of cylinder A (Fig. 3e). Therefore, ApcD in both hemidiscoidal PBS could receive significantly more excitation energy than that in red algal PBS. This feature could be critical because cyanobacterial PSI does not have its own chlorophyll-based on antennae and PBS delivers a significant amount of energy to PSI in an ApcD-dependent fashion under a state 2 condition[19]. The role of ApcD in EET to PSI in red algae is less clear because an *apcD* mutant is not available and PSI has its own chlorophyll–protein complex as light-harvesting antennae[34].

A very interesting phenomenon of the cyanobacterial PBS is that some of the cyanobacterial PBS could form arrays in situ by stacking against each other on thylakoid membranes as demonstrated by cryo-electron tomography[35]. To calculate if there is the possibility that an inter-PBS EET among those stacked PBS could occur, we docked our phycobilisomes from *Synechococcus* 7002. It was found that the cores from individual PBS are in fact perfectly aligned with each other, virtually forming very long core cylinders (Fig. 4a, b). And we also found that the presence of the first layers of the bottom cylinders is necessary for establishing close contact to the fourth layer of the adjacent cores and the distance of bilin pairs from these two layers is under 40 Å. Therefore, it is predicted that under certain conditions two adjacent hemidiscoidal PBS complexes could allow energy transfer in an inter-PBS manner (Fig. 4c, d), thus creating a complex and highly interwind energy-transfer network on the thylakoid membrane. It is also possible that the formation of arrays of PBS on thylakoid membranes could facilitate interaction between FNR located at the distal ends of the rods and stabilize FNR on PBS.

**Bilin environment provided by PBP and linkers**. Although the above energy-transfer path prediction based on the shortest distances among the bilins provides a general map of EET in the PBS cores, it is not completely understood how PBS has an EET efficiency of the near unit while the average distance between the bilin pairs in PBS is over 30 Å[7,33]. While the importance of PBP residues in the determination of bilin's energy level in PBS has been well-recognized[5–7,11,12,36,37], direct evidence for the involvement of the residues in the surrounding of the bilins in EET of PBS is scarce. In this study, the roles of the bilin-surrounding residues of ApcD and ApcF in EET were studied. The function of ApcD in state 2 to state 1 transition could be monitored by PSII fluorescence rise when dark-adapted cells were illuminated with actinic light in the presence of PSII inhibitor DCMU (Fig. 5b).

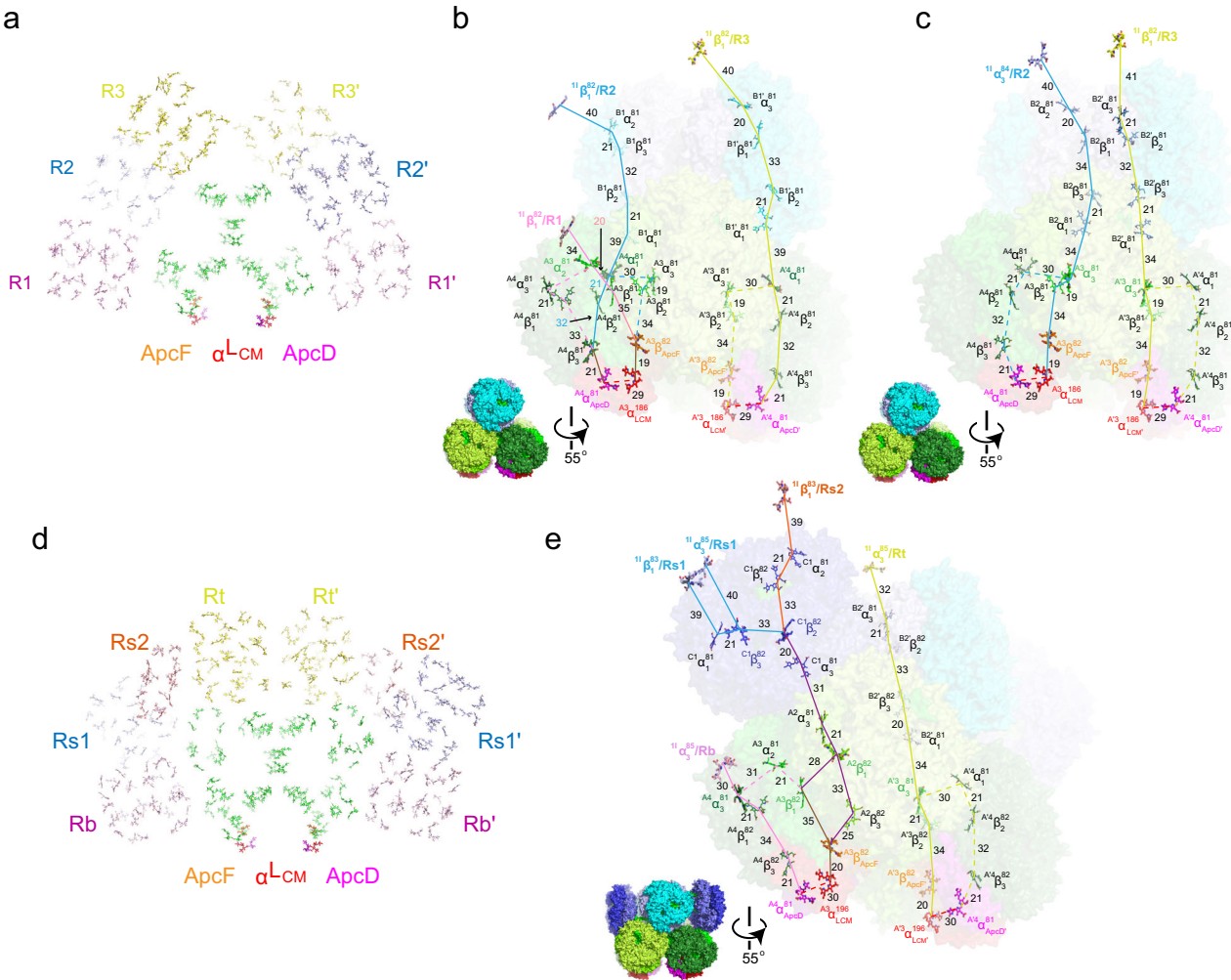

**Fig. 3 Key bilins and possible pathways for energy transfer in the core. a** Bilin distribution in the PBS from *Synechococcus* 7002. All bilins are shown in stick representation and color-coded according to their spatial locations. **b, c** Side view of the core with possible energy-transfer pathways in the PBS of *Synechococcus* 7002. Different core layers are shown in different colors. Bilins in $\alpha^{LCM}$ ($\alpha^{LCM'}$), ApcD (ApcD'), and ApcF (ApcF') are colored red, magenta, and orange, respectively. Bilins in rods are painted the same colors as the rods, in which they are located according to the coloring scheme in Fig. 1a. Bilins are shown in stick representation, and key bilins are highlighted in thicker sticks. The solid lines suggest a possible energy-transfer pathway and the dashed lines provide alternative energy-transfer pathways. The numbers indicate the distances (Å) between bilin pairs. **d** Bilin distribution in the PBS from *Anabaena* 7120. **e,** Side view of the core with possible energy-transfer pathways in the PBS of *Anabaena* 7120.

Upon illumination, the PSII fluorescence in the wild-type (WT) cell increases and reaches to plateau (state 1) in a few seconds[19]. The PSII fluorescence rise could be completely prevented by cytochrome $b_6f$ (Cyt $b_6f$) complex inhibitor DBMIB. In an *apcD* mutant, energy from PBS is constantly transferred to PSII and the PSII fluorescence remained high even in the presence of DBMIB (Fig. 5c). While a WT *apcD* gene could complement the *apcD* mutant[19], a Y88A/*apcD* gene could only partially restore the state transitions (Fig. 5d). Tyr88 is 5 Å from the ring-D of the bilin of ApcD ($\alpha_{ApcD}$) and it could form a $\pi$–$\pi$ interaction with the bilin (Fig. 5a). Another mutant *apcD* gene with a Y116A mutation completely lost the ability to restore state transitions (Fig. 5e). Y116 is located between $\alpha_{ApcD}$ and the bilin of ApcB$^{A4-II}$ ($\beta_{ApcB}^{A4-II}$) and its distance to $\alpha_{ApcD}$ is 5.0 Å. It could also form a $\pi$–$\pi$ bond with the bilin (Fig. 5a). Two other Tyr residues (Y78 and Y62) near $\alpha_{ApcD}$ and $\beta_{ApcB}^{A4-II}$ are provided by the ApcB$^{A4-II}$ and they are within 10.0 Å from these bilins. Because ApcF is a variant of ApcB and these two residues on ApcF (Phe79 and Phe60) are conserved, their roles in EET could be studied with an *apcF* mutant. The residues Phe79 and Phe60 are located between the bilin of ApcF ($\beta_{ApcF}$) and $\alpha_{ApcE}$ on the trimers A3/

A'3. Their distances to the $\alpha_{ApcE}$ are both within 5.0 Å, capable of forming $\pi$-$\pi$ interaction with the bilin (Fig. 5f)[11]. Deletion of the *apcF* gene led to a significant impairment of EET to PSII in *Synechococcus* 7002 as revealed by the 77 K fluorescence emission spectrum (Fig. 5g, h). A copy of WT *apcF* gene could fully complement this deletion mutant (Fig. 5h); but a copy of an F60A/*apcF* or an F79A/*apcF* only partially restored the EET to PSII. Consistently, a mutant *apcF* gene with double mutation F60A/F79A resulted in a more severe decrease of PSII fluorescence emission than a single mutation (Fig. 5i). Altogether, our data suggest that these aromatic residues are important for efficient EET. Another conserved residue that is within 5 Å from $\beta_{ApcF}$ is Arg77, which could form a cationic $\pi$ interaction with the bilin[12]. Mutants of ApcF with either an R77K or an R77A substitution resulted in a low PSII emission (Fig. 5j), suggesting that the guanidine group of Arg77 is needed for optimal EET to PSII.

We further examined the spatial distribution of those aromatic residues that could interact with bilins and have a global influence on EET in PBS cores. As shown in Fig. 6a, the folding of the APC and the formation of an APC trimer created a spatial ring of aromatic residues in the APC trimer and it could potentially

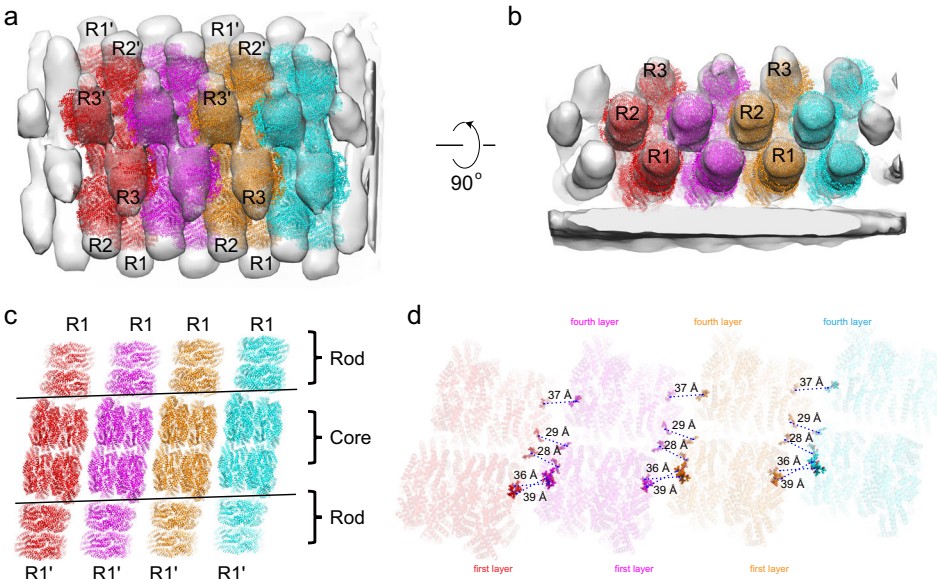

**Fig. 4 Possible energy-transfer pathways of in situ cyanobacterial PBS arrays. a, b** Structural fitting of the model of *Synechococcus* 7002 PBS into the in situ structure of the PBS array from *Synechocystis* sp. PCC 6803 (EMD-4602)[35], showing in the top (**a**) and side (**b**) views. Each PBS unit is painted in the same color. **c** Bottom view of the fitted models, highlighting that the two basal cylinders of the cores are well aligned and virtually form long cylinders. **d** Bottom view of bilin distribution of the PBS cores in (**c**). The outmost bilins from either layers 1 or layers 4 are highlighted with their respective colors in (**c**). The distances between the bilins of the first layers and the fourth layers of the adjacent PBS cores are shown.

facilitate a long-distance coupling in EET. The REP domains of ApcE, which contain Pfam00427 domains and are located within the APC hexamers, also make a significant contribution of aromatic residues. As shown in Fig. 6b, the Pfam00427 domain of the REP1 within the hexamer formed by the trimers A3 and A4 is rich in aromatic residues, and these residues are distributed evenly within the cavity. These aromatic residues together with those from APC trimers form a network of aromatic residues that could interact with the bilins of APC and with each other, therefore facilitating EET within the core (Supplementary Movie 1). This suggestion is consistent with the observation that PBS core functions as a unit in EET[22,33]. Pfam00427 domain can also be found in the linker proteins CpcG ($L_{RC}$) and CpcC ($L_R$), located within the cavities of core-proximal and core-distal hexamers of the rods, respectively. While there are several universally conserved aromatic residues in Pfam00427 domain, the Pfam00427 domain from CpcG has its own conserved aromatic residues (Supplementary Fig. 11). The distribution of aromatic residues of CpcG's Pfam00427 is polarized in such a way that they are more concentrated around the bilin of the hexamer closest to the core (Fig. 6c). The extensive interactions among this bilin and those aromatic residues could facilitate EET by decreasing the energy level of the bilin such that it becomes an energy trap of the rods, resulting in less random walk in EET and a unidirectional transfer of excitons to the core (Supplementary Movie 2). The Pfam00427 domains of $L_R$ in the core-distal hexamers have slightly fewer aromatic residues that are distributed more evenly in the cavities (Fig. 6d), which could also facilitate EET in the rods (Supplementary Movie 3). It is interesting to note that those aromatic residues are mostly Tyr and Phe residues while Trp residues are rarely present in the cyanobacterial PBPs.

Together, our work provides strong evidence that individual amino acid residues could directly participate in PBS EET. It offers a structural framework to understand why EET in PBS is unidirectional[38,39] and the importance of the linker proteins in ultrafast EET[40–42]. The structural insights would also be important to the design of artificial light-capture systems in solar energy utilization in the future.

## Methods

**Strains and growth conditions**. *Synechococcus sp*. PCC 7002 and *Anabaena sp*. PCC 7120 cells were cultured in A-plus and BG11 liquid media, respectively, bubbled with 1% $CO_2$. *Synechococcus sp*. PCC 7002-△Lr was constructed following procedures described in the previous work[28]. The *apcD* mutant was prepared as previously described[19]. To construct Y88A/Y116A mutants, *apcD* flank sequences were amplified by PCR, using the total genomic DNA as a template. The 1.5 kb upstream sequence was amplified by primer pairs of PapcD L and Y88A R/Y116A R, Y88A L/Y116A L and apcDem R (Supplementary Table 1), with site mutations on Y88 and Y116. The 1-kb downstream sequence was amplified by primers of em R L and apcDR R. A resistant gene Em$^r$ was amplified by apcDem L and em R R. The above sequences were ligated by Gibson assembly. The resulting sequences were transformed into the *apcD* mutant by conjugation. *apcF* mutant, *apcF* complemented mutant and F60A/F79A/F6979A/R77K/R77A mutants were prepared in the same way as *apcD* mutant.

**Preparation of phycobilisomes**. The PBS of *Synechococcus* 7002, *Anabaena* 7120, and *Synechococcus* 7002-ΔLr were prepared according to previous work[15] with some modifications. All operations were performed at 22 °C. Aliquots (1 g each) of the cell pellets were suspended in 8 ml Buffer B (0.75 M K/Na-PO4 buffer with 10 mM EDTA, pH 7.0) with 1 mM phenylmethylsulfonyl fluoride. Samples were disrupted through a French Press at 20,000 psi three times. Undisrupted cells and debris were removed by centrifugation at 3000×g for 10 min. Triton X-100 (20% (v/v)) was added to the supernatant to a final concentration of 2%. The mixture was gently shaken at room temperature for 30 min, and then centrifuged at 20,000×g. The middle blue-violet solution was loaded to the top of sucrose density gradients. The sucrose gradients were made from buffer B by adding sucrose to these concentrations: 0.25, 0.4, 0.55, 0.7, 0.85, and 1.0 M. Sucrose solution at each concentration was then manually added to a centrifuge tube layer by layer with the volume of 1.5, 1.5, 1.5, 2, 2, and 2 ml, respectively. The samples were centrifuged at 160,000×g for 6 h using a Ti-40 rotor on Beckman Optima L-80XP centrifuge. The sucrose of purified intact PBSs was removed by ultrafiltration with 30 KD Millipore centrifugal filters.

**Room-temperature fluorescence induction**. Fluorescence spectra from the cyanobacterial cultures were obtained following the methods in the previous work[19]. Cells were incubated in the dark for 5 min in the presence of 10 μM 3(3,4-dichlorophenyl)-1,1-dimethyl urea (DCMU) and 10 μM 2,5-dibromo-3-methyl-6-isopropylbenzoquinone (DBMIB) before the actinic light was switched on. The spectra were measured with a Nanolog FL3-2iHR fluorometer.

**77 K fluorescence emission**. Fluorescence emission spectra at 77 K were obtained following the method described previously[43]. In brief, cells were dark-adapted for 5 min to generate state 2 conditions prior to freezing in liquid nitrogen.

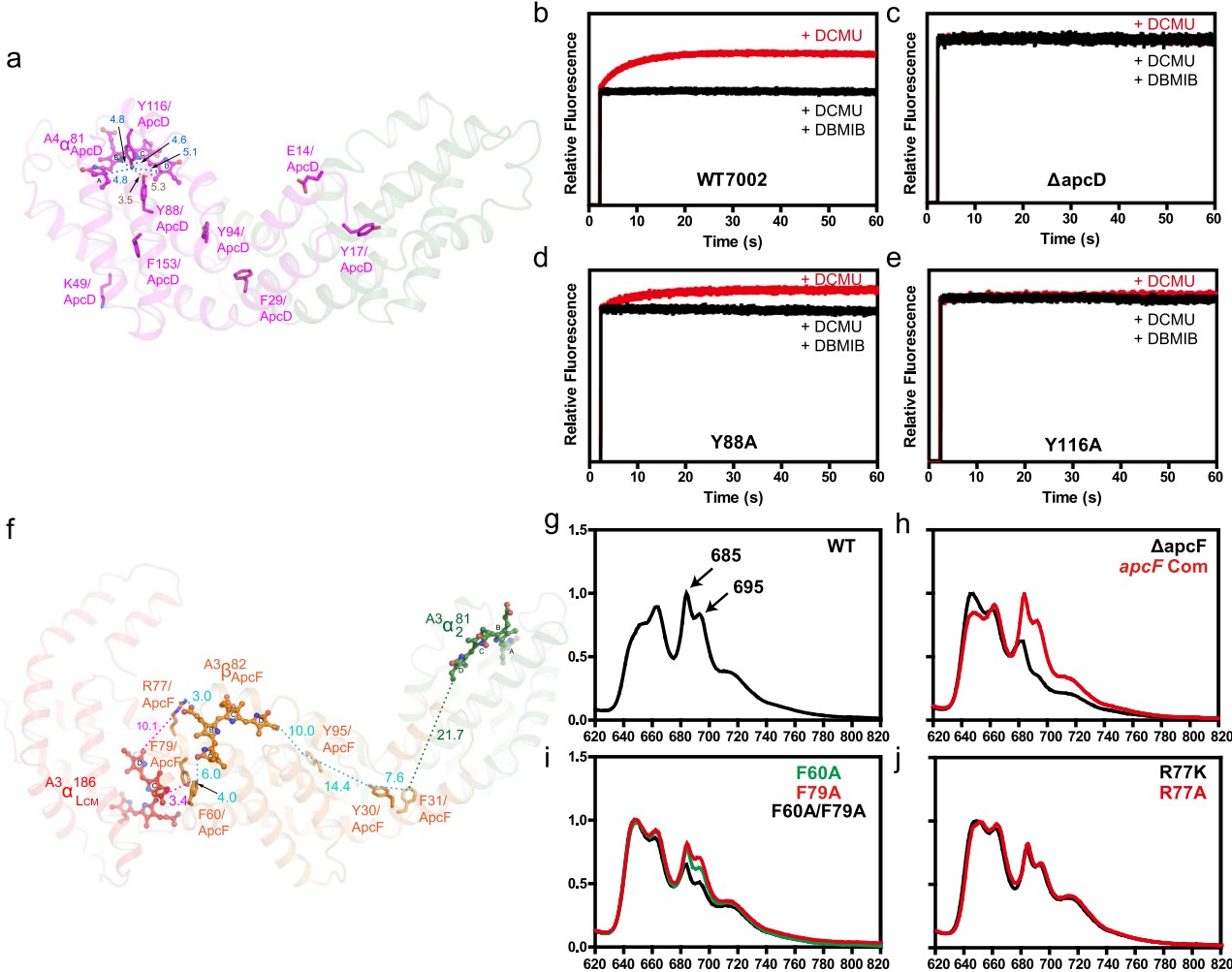

**Fig. 5 Key residues of PBP involved in energy transfer in the PBS of *Synechococcus* 7002. a** Structure of ApcD/ApcB heterodimer. ApcD and ApcB are colored red and forest green, respectively. Some important aromatic residues of ApcD are shown. Only the bilin of ApcD is shown with its ring-A through ring-D indicated. The distances of the aromatic residues near the bilin are indicated. **b–e** Fluorescence inductions in the presence of DCMU (red) or DCMU/DBMIB (black) of the wild type (**b**), the *apcD* mutant (**c**), Y88A/*apcD* mutant (**d**), and Y116A/*apcD* mutant (**e**). **f** Structure of ApcA/ApcF heterodimer. ApcF and ApcA are colored orange and forest green, respectively. The ApcA domain of ApcE is shown on the left in red. The bilins of ApcF (orange), ApcE (red), and ApcA (forest green) are shown with its ring-A through ring-D indicated. Key residues are shown, and the distances among them and to the bilins are indicated. **g–j** 77 K fluorescence emission spectra excited with a PBS absorbing light (590 nm). The wild-type (**g**); the *apcF* mutant (black curve) and its complementary strain (apcF-com, red curve) (**h**); F66A/*apcF* mutant (green curve), F79A/*apcF* mutant (red curve), and F66A-F79A/*apcF* mutant (black curve) (**i**); R77K/*apcF* (black curve) and R77A/*apcF* (red curve) (**j**). For clarity, PSII emission peaks at 685 nm and 695 nm are only indicated in (**g**) by arrows.

Fluorescence emission spectra were collected using a Nanolog FL3-2iHR fluorometer with the excitation wavelengths at 590 nm for excitation of PBS.

**Cryo-EM sample preparation and data collection.** Before preparing grids for cryo-EM, samples were concentrated to ~18 mg/ml and cross-linked using glutaraldehyde (0.005% for 7120 and 7002-ΔcpcC, 0.0125% for 7002) for 2 min to reduce the complex disassembly. Aliquots (4 μl) of the PBS samples were loaded onto one side of glow-discharged (30 s) holey Au grids (Quantifoil 1.2/1.3) and waited for 45 s. We then added twice 4-μl aliquot of a buffer (50 mM Tris pH 7.0, 0.075% NP-40) on the opposite side of the grid immediately before vitrification (to minimize preferred orientation and to dilute the high concentration of phosphate salts). After blotting for 2.5 s, the grid was plunged into liquid ethane with an FEI Vitrobot Mark IV (18 °C and 100% humidity). Cryo-grids were first screened in a Talos Arctica operated at 200 kV (equipped with an FEI CETA camera). Grids of good quality were transferred to an FEI Titan Krios operated at 300 kV with a Gatan K2 (GIF) direct electron detector for data collection using SerialEM[44]. Micrographs were collected at a nominal magnification of ×130,000 and the defocus range was set between −1.1 and −1.6 μm. Each image stack contained 32 frames and the dose rate was about 8 counts per physical pixel per second with a total exposure time of 8 s. The final calibrated pixel size at the objective scale is 1.055 Å (7120 and 7002) or 1.052 Å (7002-ΔLr).

**Image processing.** Movie stacks were first processed using MotionCor2[45] for drift correction and electron-dose weighting. The contrast transfer function (CTF) parameters of dose-weighted motion-corrected images were estimated by the program Gctf[46]. Particle picking, 2D and 3D classification, 3D auto-refinement were all done with RELION-3.1-beta[47] on twofold binned datasets unless otherwise stated.

For PBS-7002, a total of 10,030 raw movie stacks were split in two half-sets and separately processed (Supplementary Fig. 3). Around 5000 particles were manually picked to generate a set of two-dimensional (2D) templates for particle auto-picking. With these templates, 522,036 particles (from both half-sets) were auto-picked from dose-weighted images and subjected to two rounds of reference-free 2D classification. Bad and ambiguous 2D classes were discarded. The initial 3D reference was generated using RELION. For each half-set, particles were further cleaned up through two rounds of 3D classification. A final dataset, comprising 64,268 particles from both half-sets, was pooled for 3D refinement. To improve the performance of 3D refinement, particles were re-centered and re-extracted from dose-weighted images and a soft-edged mask was applied during 3D auto-refinement, leading to a 4.2-Å density map for the PBS-7002 complex based on gold-standard Fourier shell correlation (FSC) 0.143 criteria. Subsequently, the final particles were subjected to CTF refinement and Bayesian polishing, yielding an improved map at a resolution of 3.7 Å. Application of C2 symmetry led to a further improvement of the overall resolution to 3.5 Å. Mask-based refinement on the region of the PBS core could boost the resolution of the core region to 3.2 Å. To

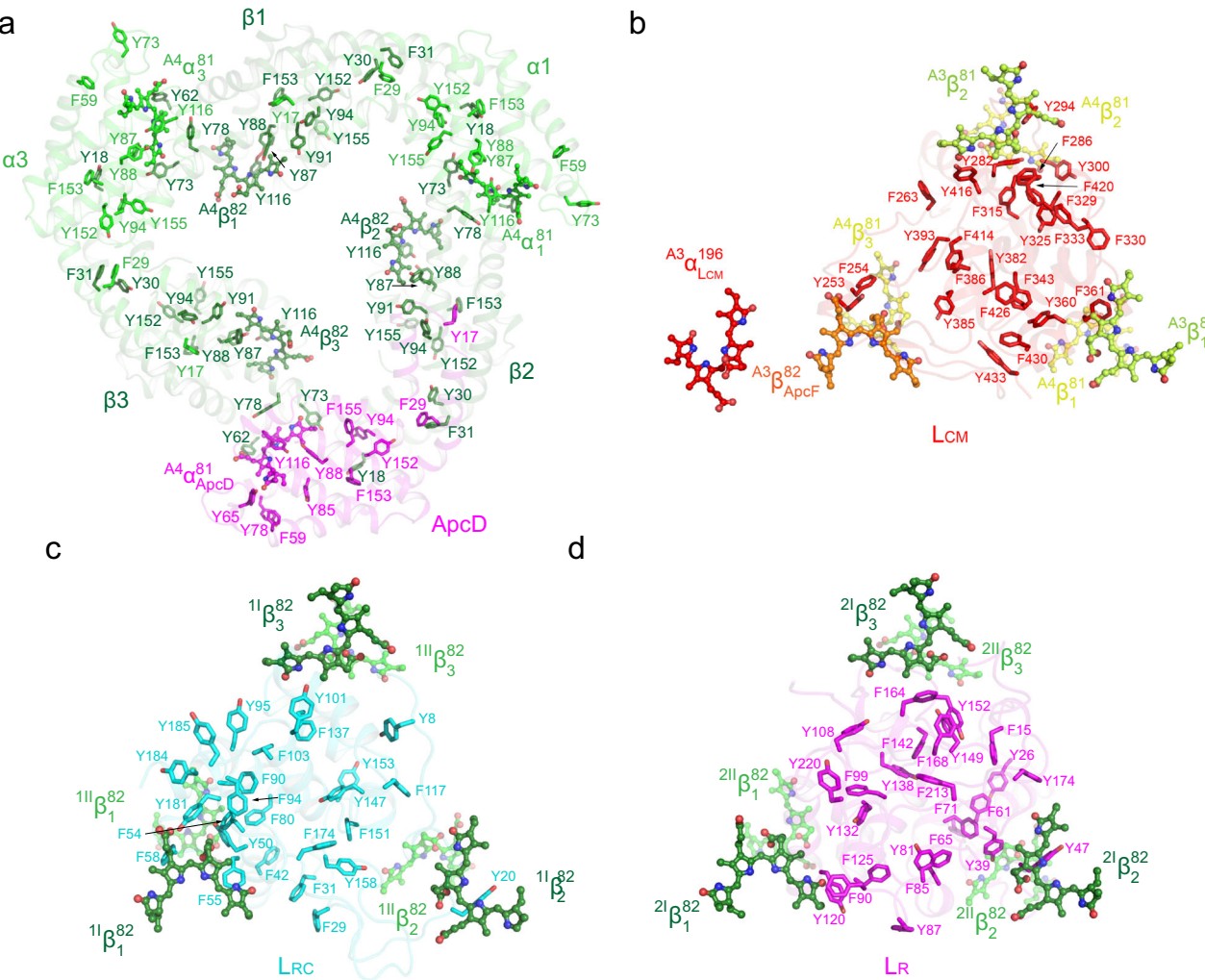

**Fig. 6 The spatial distribution of aromatic residues. a** The spatial distribution of aromatic residues in layer A4 of cylinder A. The ApcD, ApcA, and ApcB are shown in magenta, green, and forest green, respectively. **b** The spatial distribution of aromatic residues of the Pfam00427 domain in the REP1 region of ApcE. This domain is located within the cavity of the hexamer of layers A3/A4. Only the inner bilins from ApcB and ApcF (orange) plus the bilin from ApcE (red) are shown. See also Supplementary Video 1. **c, d** The spatial distribution of aromatic residues of the Pfam00427 domains of CpcG (**g**) in the core-proximal hexamer and of CpcC (**h**) in the core-distal hexamer, respectively. Only the inner bilins from CpcB subunits are shown. See also Supplementary Videos 2 and 3.

facilitate the model building of the rods, masked-based density subtraction, 3D classification, and refinement were performed on different regions of the peripheral rods, resulting in much-improved density maps for the rod terminal hexamers (3.5 Å, 4.0 Å, and 4.2 Å for R1, R2, and R3, respectively). The post-processing procedures were carried out in RELION with the modulation transfer function (MTF) of the detector and an automatically estimated B-factor applied. All the local resolution maps were calculated by the program ResMap[48].

For PBS-7120, a similar data processing workflow was used. A total of 321,385 particles were auto-picked from 13,456 raw micrographs. After several rounds of 2D and 3D classification, a final set of 62,439 particles were kept for 3D auto-refinement, resulting in a density map at a global resolution of 4.2 Å Application of CTF refinement and Bayesian polishing improved the resolution to 4.0 Å. By imposing C2 symmetry, the map could be further improved to 3.9 Å. Mask-based refinement on the PBS core led to a further improved map for the core region at 3.5 Å. Similarly, multiple soft-edged local masks were generated for different rods, and mask-based density subtraction was applied. Further 3D classification and refinement using modified particles generated a set of improved density maps at a resolution of 3.8 Å to 4.4 Å (Rb, 3.8 Å; Rs1, 4.0 Å; Rs2, 4.4 Å; Rt, 4.0 Å). For PBS-7002ΔLr, 351,181 particles were selected from 3,625 raw images and 70,637 particles were kept for final refinement. The final map was resolved at a global resolution of 3.7 Å and the core region is 3.5 Å without imposing symmetry.

**Model building and refinement.** Cryo-EM structure of the PBS from *Porphyridium purpureum* (PDB code: 6KGX) was used as the initial template for model building of 7002 and 7120 PBS complexes. The atomic models were first docked into respective density maps using UCSF chimera[49]. Sequence alignment of PBS subunits from cyanobacteria and red algae was performed using Clustal W[50]. Residue substitutions and model adjustment (rebuilding) were done manually using COOT[51]. After the manual modeling, models of individual parts were refined against their respective maps using PHENIX[52] (phenix.real_space_refine) with secondary structure and geometry restraints applied. After refinements, the model to map FSC curves showed that the consistencies between the model and the map are all between 3 and 4 Å at FSC 0.5 cutoff (with the CC value between 0.65 and 0.85). The atomic model for the overall PBS complex was derived by a combination of these refined models of individual parts. The final model was subject to another round of real-space refinement using PHENIX with secondary structure, reference model and geometry restraints applied. The final atomic models were evaluated using Molprobity[53] and the statistics of data collection and model validation were included in Supplementary Table 2.

**Reporting summary.** Further information on experimental design is available in the Nature Research Reporting Summary linked to this paper.

## Data availability
Atomic models of the PBS structures from *Synechococcus* 7002 and *Anabaena* 7120 have been deposited in the Protein Data Bank (PDB) under accession number 7EXT (*Synechococcus* 7002) and 7EYD (*Anabaena* 7120), respectively. The cryo-EM from *Synechococcus* 7002, *Anabaena* 7120, and *Synechococcus* 7002-ΔLr maps have been deposited in the Electron Microscopy Data Bank with the accession code EMD-31373

(*Synechococcus* 7002), EMD-31381 (*Anabaena* 7120), and EMD-31483 (*Synechococcus* 7002-ΔLr), respectively. All other data are available from the corresponding authors.

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

## Acknowledgements

We would like to dedicate this article to Professor Alexander N. Glazer, who had many pioneering contributions to the study of phycobilisomes and their applications. We thank the Core Facilities of Peking University School of Life Sciences for assistance with negative-staining electron microscopy; the Cryo-EM Platform and the Electron

Microscopy Laboratory of Peking University for cryo-EM data collection; the High-performance Computing Platform of Peking University for help with computation. We also thank the National Center for Protein Sciences at Peking University for technical support. This work was supported by the National Science Foundation of China (31725007 and 31630087 to N.G., 32070203 and 91851118 to J.Z., 31800197 to Z.Z.), the Ministry of Science and Technology of China (2019YFA0508904 to N.G., 2017YFA0503703 to J.Z.), and the Qidong-SLS Innovation Fund to N.G.

## Author contributions

N.G. and J.Z. conceived the project; Z.Z. purified PBS samples; L. Z. and G.W. collected cryo-EM data; L.Z. and G.W. processed the cryo-EM data; X.L., K.Z., and P.W. conducted biochemical analysis; L.Z., N.G., and J.Z. wrote the manuscript; all authors discussed and commented on the results and approved the manuscript.

## Competing interests

The authors declare no competing interests.
