## [Peer Review File · Nature Communications]

Structural Insight into the Mechanism of Energy Transfer in the Cyanobacterial PhycobilisomesREVIEWER COMMENTS

Reviewer #1 (Remarks to the Author):

This paper provides near-atomic resolution cryo-EM structures of phycobilisome antenna complexes from two cyanobacteria. These structures have long been anticipated and the current work fills a major gap in the understanding of energy transfer in these important systems.

Overall, I am strongly supportive of publication and have only a few comments on the work and the manuscript. The authors are world experts in these systems and I have no doubt that the work has been done carefully.

One place where the description does not go into much detail is where and how the PBS complexes attach to both PSI and PSII. This aspect has been addressed by H. Liu and co-workers (Science Advances 7: eaba5743 (2021)). I realize that the current preparations did not include the RCs, but this is the ultimate issue and I think deserves at least some discussion.

I noted with interest the presumed role of the aromatic residues in the energy transfer. However, I was very surprised that these aromatics only included Tyr and Phe, and no Trp. One would naively expect Trp to be the most effective due to its lower energy excited state compared to Tyr and Phe and the fact that it is well established as important in other photosynthetic antenna systems. Is the lack of Trp a well-known feature of these systems? I was not previously aware of this fact.

Finally, one of the most interesting features that is currently under active investigation in many laboratories is the regulatory protein orange carotenoid protein (OCP). These structures provide the opportunity to explore using modeling the binding of the OCP. I think at least some discussion of this system is warranted.

Reviewer #2 (Remarks to the Author):

In this manuscript, Gao and co-workers describe the complete three-dimensional structures of two cyanobacterial Phycobilisome antenna complexes (PBS) – from *Anabaena* sp. PCC 7120 (at 3.5Å resolution) and *Synechococcus* sp. PCC 7002 (at 3.9Å resolution), using single particle reconstruction (SPR) by cryo-TEM. These are the first cyanobacterial structures of the PBS at this resolution, joining the two existing structures obtained by the same method from two red algal species. While of lower resolution than those of the red algae, these two new structures actually represent the more common hemi-discoidal PBS that has been the basis for almost all experimental studies on the PBS for over 50 years. The group that provided the red-algal structures noted in their first paper that obtaining cyanobacterial structures was problematic due to lack of stability that leads to partial (and stochastic) disassembly of the PBS, preventing efficient SPR. Thus, the study submitted by this team here have managed to overcome (at least partially) this problem. The two structures are described in proper detail in the paper, including detailed descriptions of the core, the rods (which are less ordered), and the internal linker proteins. Potential excitation energy transfer (EET) pathways, based on relative phycobilin positions are suggested. In addition, the authors performed a series of fluorescence based (RT and 77K) measurements of energy transfer from the PBS to PSII in a series of mutants, to identify specific amino acid residues that affect EET. As can be expected, aromatic residues play an important role in EET, but the authors also add important observations on other residues, such as Arginines. One other section was the attempt to obtain the position of bound FNR. This protein is known to associate at the distal ends of rods, was found in the isolated complexes – however could not be identified in either WT PBS, or in a mutant that contains only a single hexamer in each rod position. This last experiment led to an additional cryo-EM structure that has not been completed to the level required for deposition. In general, this shows that FNR binding to isolated complexes is quite weak. There are a few issues that should be rectified before acceptance, however in general this is an important paper and will be of significant interest to the general readership.

1. Based on the material provided, the authors have not yet completed deposition into the PDB. The files provided specifically state that these are not to be used upon manuscript submission. While I have no doubt that this is the intension of the authors, the manuscript cannot be accepted without completing this task.
2. Cores with 3 or 5 cylinders should be called tri-cylindrical and penta-cylindrical cores, respectively. This is the terminology used in decades of papers on the PBS.
3. Shouldn't the different domains of the LCM be called ARM and REP domains, and not REG domains as denoted in the paper here (again, to conform to previous literature)?
4. Fig. 3. It is not clear from the legend what is the identity of the middle structure (there should be a, b, and c. notation, I suspect). Do the authors feel confident in their measurements of $\pm 0.1 \text{ \AA}$ with the resolution obtained in their SPR?
5. As seen in the negative stained images in ED Fig. 2, there is a high degree of heterogeneity in the particles. While it is convenient to choose those particles that best fit the model of the hemidisoidal PBS found in the literature, can the authors state what is the ratio of particles used in the analysis versus all other forms of the PBS? It has been shown by cryo tomography (Rast et al, Nat. Plants 2019, not cited) that the PBS forms arrays on the thylakoid membranes, with close contacts between adjacent complexes. If upon detergent driven disassociation of the PBS from the membrane, significantly more complexes are not hemidisoidal, then how can one definitely state what a "real" PBS looks like? In ED Figs. 3 and 4, there seem to be more non-regular complexes, than regular complexes. It should be stated that it has been shown that a completely non-regular PBS (as seen by both negative stained and cryo-EM) performs efficient EET (David et al. BBA 2014, not cited).
6. Could the results pertaining to the lack of stability of the FNR binding be due to similar reasons – lack of contact between adjacent complexes? Perhaps this should be mentioned.
7. The red algae PBS structures exhibited non canonical cores, unlike the structures presented here. Those cores were missing one (bottom layer cylinders) or two (to layer) APC timers, which exposes the surface of the α subunits – which normally should be covered by another layer of α subunits. The authors suggest an evolutionary reason for these differences, however I personally find this highly unlikely. Can the authors suggest any structural reason for the difference between the cyanobacterial and red algae cores? Something that makes the cyanobacterial core cylinders actually more stable than the red-algal cylinders? Could this be caused by what was asked in comment #5?
8. The authors description of the linker as playing a critical role in modifying EET in such a fashion that the different chromophores are tuned to direct the EET properly from the rods to the core and from there to the RC has been suggested in the past and indeed it was suggested that the name of these proteins should be tuning proteins. This should be cited (David JMB 2011).
9. Local density surrounding specific structural facets are quite good (ED Figs. 3 and 4). As the authors are probably aware, it has been suggested in the past that the reason for the significant red-shift in ApcD, is due to a forced planarity in the bilin chromophore (Peng et al. Acta Cryst. D 2014 – cited). Are the maps good enough to confirm this observation (or not)?
10. The animated movies are well done, and are helpful.
11. Fig. 4. Please check legend (there are typos). There are no arrows in panel Di.
12. Page 4 – concentration misspelled.

Reviewer #3 (Remarks to the Author):

The manuscript entitled "Structural Insight into the Mechanism of Energy Transfer in Cyanobacterial Phycobilisomes" represents an important advance in the field of photosynthesis as it describes the first structures of bacterial phycobilisomes. The only structures previously available were solved from eukaryotic red algae. Here, phycobilisome structures were solved from two different cyanobacterial species, *Anabaena* 7120 and *Synechococcus* 7002. Even though the resolution is not so state of the art for single particle cryo-EM (e.g compared to previously obtained red algae reconstructions), I feel that the methodology is good and the sample appears to be quite difficult to process, hence explaining the difficulty to reach higher resolution. The manuscript describes in details the architecture and composition of these molecular complexes, notably how the core subunits and peripheral rods are attached together via linker proteins. Additionally, the authors propose a model to explain the process of energy transfer in phycobilisomes. While this represents an important result, I feel the

manuscript would benefit from several changes to be more easily understood by a wider audience. Notably, the manuscript heavily rely on the fact that the reader would have some prior knowledge of phycobilisomes and often compares the cyanobacterial structures with the previously solved red algae structures, without actually comparing them in e.g a figure.

Major comments:

A - Results – “Negative-staining electron microscopy (nsEM) analysis revealed a large variation in the length of peripheral rods” Is it possible to quantify it? What’s the average number of unit per peripheral rods?

B - For each models please provide the map to model correlation curves and numbers (output from phenix)

C – Overall, I found myself going a lot back and forth between the figures. This may be due to the fact that everything has been condensed in four main figures. Here’s some propositions that I think would improve readability:

-In figure 1: There’s a lot happening in this figure. First, I would align on the same line a, c and f as well as g, i and l. I would remove b and h (2D classes) and put them in supp. As a replacement I would illustrate the fact that there’s additional rod units with dashed-lines in either a or f (similar in g or l). In a, d, g and f, I don’t understand why you would show the molecular model inside a transparent density. We cannot see it anyway. Just show the density.

Personally, since you speak of the billins and energy transfer in figure 3, I would move e and k to figure 3. That way, you could slightly enlarge panels a, d, g and f.

-In figure 2 you should show densities. Models are nice but it would be better to see the actual data, notably in panels a, f, g and h. Especially since the representative densities showed in supp are mostly from the core subunits where the resolution is the highest, and may not represent the rest of the complex.

Also, the article would really benefit from an additional figure comparing the cyanobacterial and eukaryotic phycobilisomes, since the comparison is done several times in the manuscript. For example in the description of the core structure “Notably, the structure of red algal PBS contains three layers of $\alpha\beta$ trimers in the basal cylinders and two layers of $\alpha\beta$ trimers in the top cylinders” and the sentence just after “Structural comparison indicates that the equivalent layers of A1/A’1 in ...”. You are not showing any structural comparison here.

D – Results – Maybe I missed it but I wasn’t able to identify ApcA, ApcB and ApcE; where are they in the structure?

E – Results – “no Pfam01383 domain of any source was observed in the cavities of the rod hexamers from the Δ Lr-PBS (Extended data Fig. 6g).” You are not showing the WT version, hence I can only believe you. Please put the WT density side-by-side with the mutant one.

F – In Figure 4, the “i” “ii” “iii”... are misleading with other panel names, please replace them.

Minor comments:

-Please remove “near atomic resolution” from the abstract. This term has no meaning anymore in cryo-EM. State the real resolution (3.5 and 3.9) or just remove it.

-Introduction – add “about” before 2.4 billion years ago

-Introduction – The first sentence should be split in two.

-Results – “Both PBS contain phycocyanin (PC), allophycocyanin (APC) and linker proteins but lack phycoerythrin.” Is it important to precise that it lacks phycoerythrin? What is it?

-Results – In the “linker protein” paragraph, could you please explain what’s the role of FNR here. An introductory sentence may be necessary.

-In figure 4d – in the text, a R77A mutant is mentioned but not shown.

-Extended data fig.9 is not cited in the text.

-The movies are nice but would benefit from annotations.

-How does your single particle structures compare with the in situ phycobilisome structure of *Synechocystis*?

REVIEWER COMMENTS

Reviewer #1 (Remarks to the Author):

This paper provides near-atomic resolution cryo-EM structures of phycobilisome antenna complexes from two cyanobacteria. These structures have long been anticipated and the current work fills a major gap in the understanding of energy transfer in these important systems.

Overall, I am strongly supportive of publication and have only a few comments on the work and the manuscript. The authors are world experts in these systems and I have no doubt that the work has been done carefully.

One place where the description does not go into much detail is where and how the PBS complexes attach to both PSI and PSII. This aspect has been addressed by H. Liu and co-workers (Science Advances 7: eaba5743 (2021)). I realize that the current preparations did not include the RCs, but this is the ultimate issue and I think deserves at least some discussion.

We thank the reviewer for these thoughtful suggestions. We have added some discussion in the text (Page 5).

I noted with interest the presumed role of the aromatic residues in the energy transfer. However, I was very surprised that these aromatics only included Tyr and Phe, and no Trp. One would naively expect Trp to be the most effective due to its lower energy excited state compared to Tyr and Phe and the fact that it is well established as important in other photosynthetic antenna systems. Is the lack of Trp a well-known feature of these systems? I was not previously aware of this fact.

In general, all the PBS subunits of *Synechococcus* 7002 and *Anabaena* 7120 have very few Trp residues. ApcA, ApcB, ApcF and CpcB in both PBSs contain no Trp residue.

In the 7002 PBS, ApcD and CpcA only possess one Trp residue (W87 in ApcD, W128 in CpcA), far less than the numbers of Tyr and Phe residues in these two subunits. These two Trp residues are at equivalent positions of the conserved Pfam00427 domain. Mutation of ApcD-W87 (7002 PBS) into alanine or leucine had no apparent effect in the state transitions (Response Figure 1), indicating a non-essential role in EET. Trp residues in ApcE of the 7002 PBS locate at residues 154, 407 and 584 (in α Lcm, REP1, REP2) and these sites are spatially equivalent to W87 in ApcD (Pfam00427).

Similarly, in the 7120 PBS, ApcD contains two Trp residues (W59 and W87) and CpcA has one Trp residue (W129). The residue W59 is far from the bilins. Compared

with the 7002 PBS, two extra Trp residues (W769 and W1117, in REP3 and REP4, respectively) are present in ApcE of the 7120 PBS (again in equivalent positions of respective Pfam00427 domain).

Based on the reviewer's comment, we added a sentence in the text: it is interesting to note that those aromatic residues are mostly Tyr and Phe residues while Trp residues are rarely present in the cyanobacterial PBPs (page 11).

Response Figure 1| Fluorescence inductions in the presence of DCMU (red) and DCMU/DBMIB (black). Left panel shows the fluorescence inductions of W87A/*apcD* mutant and right panel shows the fluorescence inductions of W87L/*apcD* mutant.

Finally, one of the most interesting features that is currently under active investigation in many laboratories is the regulatory protein orange carotenoid protein (OCP). These structures provide the opportunity to explore using modeling the binding of the OCP. I think at least some discussion of this system is warranted.

Suggestion is well taken. We have added some discussion in the revision (Page 6).

Reviewer #2 (Remarks to the Author):

In this manuscript, Gao and co-workers describe the complete three-dimensional structures of two cyanobacterial Phycobilisome antenna complexes (PBS) – from *Anabaena* sp. PCC 7120 (at 3.5Å resolution) and *Synechococcus* sp. PCC 7002 (at 3.9Å resolution), using single particle reconstruction (SPR) by cryo-TEM. These are the first cyanobacterial structures of the PBS at this resolution, joining the two existing structures obtained by the same method from two red algal species. While of lower resolution than those of the red algae, these two new structures actually represent the more common hemi-discoidal PBS that has been the basis for almost all experimental studies on the PBS for over 50 years. The group that provided the red-algal structures noted in their first paper that obtaining cyanobacterial structures was problematic due to lack of stability that leads to partial (and stochastic) disassembly of the PBS, preventing efficient SPR. Thus, the study submitted by this team here

have managed to overcome (at least partially) this problem. The two structures are described in proper detail in the paper, including detailed descriptions of the core, the rods (which are less ordered), and the internal linker proteins. Potential excitation energy transfer (EET) pathways, based on relative phycobilin positions are suggested. In addition, the authors performed a series of fluorescence based (RT and 77K) measurements of energy transfer from the PBS to PSII in a series of mutants, to identify specific amino acid residues that affect EET. As can be expected, aromatic residues play an important role in EET, but the authors also add important observations on other residues, such as Arginines. One other section was the attempt to obtain the position of bound FNR. This protein is known to associate at the distal ends of rods, was found in the isolated complexes – however could not be identified in either WT PBS, or in a mutant that contains only a single hexamer in each rod position. This last experiment led to an additional cryo-EM structure that has not been completed to the level required for deposition. In general, this shows that FNR binding to isolated complexes is quite weak. There are a few issues that should be rectified before acceptance, however in general this is an important paper and will be of significant interest to the general readership.

1. Based on the material provided, the authors have not yet completed deposition into the PDB. The files provided specifically state that these are not to be used upon manuscript submission. While I have no doubt that this is the intention of the authors, the manuscript cannot be accepted without completing this task.

Atomic models of the PBS structures from *Synechococcus* 7002 and *Anabaena* 7120 have been deposited in the Protein Data Bank (PDB) under accession number 7EXT (*Synechococcus* 7002) and 7EYD (*Anabaena* 7120), respectively. The cryo-EM maps of PBS structures from *Synechococcus* 7002, *Anabaena* 7120 and *Synechococcus* 7002- Δ Lr have been deposited in the Electron Microscopy Data Bank with the accession code EMD-31373 (*Synechococcus* 7002), EMD-31381 (*Anabaena* 7120) and EMD-31483 (*Synechococcus* 7002- Δ Lr), respectively.

2. Cores with 3 or 5 cylinders should be called tri-cylindrical and penta-cylindrical cores, respectively. This is the terminology used in decades of papers on the PBS.

We thank the reviewer for this suggestion. We have followed this convention in the revision.

3. Shouldn't the different domains of the LCM be called ARM and REP domains, and not REG domains as denoted in the paper here (again, to conform to previous literature)?

Suggestion is well taken. We have changed them in the revision.

4. Fig. 3. It is not clear from the legend what is the identity of the middle structure (there should be a, b, and c. notation, I suspect). Do the authors feel confident in their measurements of $\pm 0.1 \text{ \AA}$ with the resolution obtained in their SPR?

Suggestions are well taken. We have revised the figures and changed the labels of measurements ($\pm 1 \text{ \AA}$).

5. As seen in the negative stained images in ED Fig. 2, there is a high degree of heterogeneity in the particles. While it is convenient to choose those particles that best fit the model of the hemidiscoidal PBS found in the literature, can the authors state what is the ratio of particles used in the analysis versus all other forms of the PBS? It has been shown by cryo tomography (Rast et al, Nat. Plants 2019, not cited) that the PBS forms arrays on the thylakoid membranes, with close contacts between adjacent complexes. If upon detergent driven disassociation of the PBS from the membrane, significantly more complexes are not hemidiscoidal, then how can one definitely state what a “real” PBS looks like? In ED Figs. 3 and 4, there seem to be more non-regular complexes, than regular complexes. It should be stated that it has been shown that a completely non-regular PBS (as seen by both negative stained and cryo-EM) performs efficient EET (David et al. BBA 2014, not cited).

The heterogeneity of PBS particles came from several sources: (1) the natural variation in the length and orientation of the rods; (2) partial dissociation of the PBS complexes during sample purification, including the Triton X-100 based solubilization and sucrose-based ultracentrifugation; (3) complex dissociation or even denaturing by surface tension at the air-water interface on the grids during plunge-freezing of cryo-EM grids. Therefore, it is difficult to isolate these effects of different sources.

As shown in Extended Data Figures 3, 4 and 7, we did not introduce knowledge-based bias in particle picking or 2D classification, all the particles picked by RELION were subjected to standard procedures of 2D classification. Although 3D classification indeed revealed structures of “non-regular” PBS complexes, the fraction of expected “normal” hemidiscoidal complexes is the largest for all the samples (ED figure 3, 32% for the first round of 3D classification; ED figure 4, 41% for the first round; ED figure 7, 50% for the first round). Moreover, these less populated non-regular structures from different 3D classes exhibit the largest variation at peripheral rods (also evident from 2D average images) (Extended Data Figure 3). Of note, a few 3D classes (Extended Data Fig. 3, Extended Data Fig 7d, lower right panel) appear to have unstable core cylinders. But these classes are in the least populated groups and the maps are of very low resolution. Therefore, they should represent those partially “damaged” particles during cryo-grid freezing. In summary, we believe that most PBS complexes in our study should still be hemidiscoidal.

The beautiful *in situ* structure of PBS arrays by cryo-electron tomography (Rast et al, Nat. Plants 2019) showed that hemidiscoidal PBS complexes could closely stack against each other to form regular arrays on the thylakoid membranes. Importantly, although these arrays display distinct orientations on the same membrane, they are all formed by similar hemidiscoidal units. This indicates that a hemidiscoidal PBS complex is the basic unit of PBS array assembly *in vivo*. The *in situ* structure also explains the advantage of a general hemidiscoidal shape for the PBS to form regular arrays: two adjacent hemidiscoidal PBS could stack very close to each other, potentially allowing inter-PBS energy transfer.

Response Figure 2| Segmented surfaces of different membrane regions decorated with PBS (yellow) (adapted from Rast et al, Nat. Plants 2019).

David and co-workers showed that PBS complexes purified from *Thermosynechococcus vulcanus* (TvPBS) display non-regular structures from both nsEM and cryo-EM. It has to be noted that the 3D structure presented in this study was of very low resolution (30 Å), and the authors were unable to assign either PC or APC hexamers into the map. From Figure 6 of this study, it is highly likely that the reconstructed map also represents a stable core with partially dissociated rods.

We appreciate the reviewer for raising this issue. We have cited these references and added relevant discussion in the revision.

6. Could the results pertaining to the lack of stability of the FNR binding be due to similar reasons – lack of contact between adjacent complexes? Perhaps this should be mentioned.

The FNR contains three conserved domains (CpcD-like domain, FAD binding domain and NADP-binding domain). The CpcD-like domain should be located in the cavities of the distal $\alpha\beta$ hexamers. The other two domains should be in protruding position on the terminal PC hexamer. While the interaction between FNR could play a role in stabilizing FNR in PBS when PBS are in arrays, FNR is stably present in PBS since individual PBS isolated in high phosphate buffer contain FNR. The instability of FNR-binding to PBS under cryo-EM condition could be due to some other reasons, such as disassociation of FNR from PBS at low temperature.

Following the reviewer's suggestion, we also added a sentence in the revision (page 10)

7. The red algae PBS structures exhibited non canonical cores, unlike the structures presented here. Those cores were missing one (bottom layer cylinders) or two (top layer) APC trimers, which exposes the surface of the α subunits – which normally should be covered by another layer of α subunits. The authors suggest an evolutionary reason for these differences, however I personally find this highly unlikely. Can the authors suggest any structural reason for the difference between the cyanobacterial and red algae cores? Something that makes the cyanobacterial core cylinders actually more stable than the red-algal cylinders? Could this be caused by what was asked in comment #5?

The suggestion was first brought up by the authors of the paper describing red algal PBS (Nature, 2017 by Zhang et al, reference 11), which wrote “This core structure is likely to be evolutionarily derived from the core of hemidisoidal PBS by the elimination of the exterior trimers of all three cylinders except for those with terminal emitters. The formation of a more compact core structure leads to the loss of 24 PCBs in the core and it could be an adaptation to habitats in which the red light absorbed by PCBs is limited.” We think the suggestion is scientifically sound and cited it in our manuscript. To avoid any ambiguity, we removed the word “evolutionarily” in our revised manuscript.

As for the reviewer's question about structural reason for the difference between cyanobacterial and red algal cores, we think that four trimer layers in one cylinder are more stable in hemidisoidal cores. The removal of the outside layers of the core trimers in red algal PBS core, on one hand, reduces the number of PCB, which is needed mostly for energy transfer process while not so much for their light absorption function. On the other hand, this omission provides space for attaching four more rods, Rod d, d', e, and e' (Response Fig. 3).

In the context of PBS array *in vivo*, cyanobacterial PBS complexes would stack very close one by one, and the cores from individual PBS are in fact perfectly aligned with each other, virtually forming very long core cylinders. The presence of the fourth

layer is necessary for establishing close contact between adjacent cores, which might allow the energy transfer between them. Because this is a critical feature of cyanobacterial PBS, we added a figure (Fig. 4) to describe it. In contrast, the red algal PBS contains more rods on the hemidiscoidal face, which would apparently prevent the close contact between cores from adjacent PBS complexes in its high-order structure on the membrane.

Response Figure 3| Schematic diagram showing the organization of the rods and core cylinders from three perpendicular views. (adapted from Zhang et al, Nature 2017).

8. The authors description of the linker as playing a critical role in modifying EET in such a fashion that the different chromophores are tuned to direct the EET properly from the rods to the core and from there to the RC has been suggested in the past and indeed it was suggested that the name of these proteins should be tuning proteins. This should be cited (David JMB 2011).

Suggestion is well taken. We have cited it in the text.

9. Local density surrounding specific structural facets are quite good (ED Figs. 3 and 4). As the authors are probably aware, it has been suggested in the past that the reason for the significant red-shift in ApcD, is due to a forced planarity in the bilin chromophore (Peng et al. Acta Cryst. D 2014 – cited). Are the maps good enough to confirm this observation (or not)?

The resolutions of the core regions of the 7002 and 7120 PBS maps are 3.2 Å and 3.5 Å, respectively. We have carefully checked the local densities of the bilins in the ApcA, ApcB and ApcD in these two maps. As in the 1.7-Å high-resolution crystal structure (Peng et al. Acta Cryst. D 2014), the PCB of ApcD in our structures adopts a similar configuration: the rings of B, C and D are in a co-planar geometry. Notably, PCBs of ApcD in red algae PBS structures (Zhang, J. et al. Nature 2017; Ma, J.F. et al. Nature 2020) are also in a planar configuration.

We also noticed a slight deviation of the ring D in ApcA-PCB. Nevertheless, the accurate interpretation of subtle structural differences of bilin planarity would require structures solved at below 2-Å resolution.

Response Figure 4| Structural comparison of PCB in different subunits. a, Local density of a representative PCB in ApcA. b, Local density of a representative PCB in ApcB. c, Local density of a representative PCB in ApcD. d, Structural comparison of PCB in ApcA, ApcB and ApcD.

10. The animated movies are well done, and are helpful.

Thank you.

11. Fig. 4. Please check legend (there are typos). There are no arrows in panel Di.

Thank you. We have added arrows in panel Di (Figure 5g).

12. Page 4 – concentration misspelled.

We thank the reviewer for catching this error. It has been corrected in the revision.

Reviewer #3 (Remarks to the Author):

The manuscript entitled “Structural Insight into the Mechanism of Energy Transfer in Cyanobacterial Phycobilisomes” represents an important advance in the field of photosynthesis as it describes the first structures of bacterial phycobilisomes. The only structures previously available were solved from eukaryotic red algae. Here, phycobilisome structures were solved from two different cyanobacterial species, *Anabaena* 7120 and *Synechococcus* 7002. Even though the resolution is not so state of the art for single particle cryo-EM (e.g compared to previously obtained red algae reconstructions), I feel that the methodology is good and the sample appears to be quite difficult to process, hence explaining the difficulty to reach higher resolution. The manuscript describes in details the architecture and composition of these molecular complexes, notably how the core subunits and peripheral rods are attached together via linker proteins. Additionally, the authors propose a model to

explain the process of energy transfer in phycobilisomes. While this represents an important result, I feel the manuscript would benefit from several changes to be more easily understood by a wider audience. Notably, the manuscript heavily rely on the fact that the reader would have some prior knowledge of phycobilisomes and often compares the cyanobacterial structures with the previously solved red algae structures, without actually comparing them in e.g a figure.

Major comments:

A - Results – “Negative-staining electron microscopy (nsEM) analysis revealed a large variation in the length of peripheral rods” Is it possible to quantify it? What’s the average number of unit per peripheral rods?

The rods in the 7120 PBS are very crowded and it is very hard to count the number of PC hexamers in EM images. Even for the 7002 PBS, the very limited side-views of PBS particles have prevented an accurate analysis of the rod length. As suggested, we did some quantification on the peripheral rods based on the negative-staining EM images of the 7002 PBS. The results show that the number of PC hexamer could reach 6. Rods of R1 appear to be the longest. But, this is not necessarily true. Because R1 is the bottom rod, and could be easily seen from multiple angles.

Therefore, we think that it is not a strict statistical analysis. Also, for this reason, we intend not to show this figure in the supplementary data.

We also modified the text on page 3, emphasizing the hexamer number of PC rods could reach as large as 6.

Response Figure 5| Histogram of hexamer numbers of in peripheral rods of the 7002 PBS. The count of hexamer numbers of per peripheral rods in R1 (left panel), R2 (middle panel) and R3 (right panel) of 7002.

B - For each models please provide the map to model correlation curves and numbers (output from phenix)

As stated in the method, the high-resolution maps for different regions of the PBS were obtained through mask-based 3D classification and refinement. The modelling of different parts was done with respective maps separately. These models were then

combined to generate a final model for the overall PBS complexes. As shown in Response Figure 6, the map to model correlation curve for the overall map vs the final model is generally poor (panels a and d). This is due to the fact that peripheral rods in the overall map have very large variations (both the length and orientation). Therefore, the curve and CC value for the overall map could be misleading. We also calculated the map to model correlations for individual parts of the PBS (panels, b, c, e and f). In these curves, at FSC 0.5 cutoff, the consistencies between the model and the map for different parts are all between 3 to 4 Å, suggesting a reasonably good fit. To facilitate the modelling, the rods are in fact separated into more parts for mask-based refinement. Therefore, we think that it is probably not necessary for us to provide the map to model correlation curves and CC values for all individual parts. The figure would require more than a dozen of panels.

We have modified the methods to add more details of modelling. We also added a general description of the curves and CC values for individual parts of the PBS in the methods. Of course, we will be happy to make an additional figure if the reviewer insists.

Response Figure 6| Map to model correlation curves. a, The map to model correlation curves of the overall structure of 7120. **b,** The map to model correlation curves of the core of 7120. **c,** A representative map to model correlation curves of the rod of 7120. **d,** The map to model correlation curves of the overall structure of 7002. **e,** The map to model correlation curves of the core of 7002. **f,** A representative map to model correlation curves of the rod of 7002.

C – Overall, I found myself going a lot back and forth between the figures. This may be due to the fact that everything has been condensed in four main figures. Here’s some propositions that I think would improve readability:

Suggestion is well taken. We have changed the presentation of figures to make them more friendly for the readers.

-In figure 1: There's a lot happening in this figure. First, I would align on the same line a, c and f as well as g, i and l. I would remove b and h (2D classes) and put them in supp. As a replacement I would illustrate the fact that there's additional rod units with dashed-lines in either a or f (similar in g or l). In a, d, g and f, I don't understand why you would show the molecular model inside a transparent density. We cannot see it anyway. Just show the density.

We thank the reviewer for the thoughtful suggestions. We have changed the presentation of Figure 1 to make it clearer. We have moved panels 1b and 1h to ED fig. 3c and 4c, respectively. We also tried to use dashed-lines in panels a and f, but the figure turned out to be very ugly.

Personally, since you speak of the billins and energy transfer in figure 3, I would move e and k to figure 3. That way, you could slightly enlarge panels a, d, g and f.

We have changed them in the revision as suggested.

-In figure 2 you should show densities. Models are nice but it would be better to see the actual data, notably in panels a, f, g and h. Especially since the representative densities showed in supp are mostly from the core subunits where the resolution is the highest, and may not represent the rest of the complex.

Suggestion is well taken. We have now generated a new Extended Data Figure showing the match between actual densities and models for each PBS subunits. Panels a, f, g and h in Figure 2 focus on structural comparison of PBS subunit as a whole, and with a highlight on their differences. Superimposition of densities would make the panels very hard to read.

Response Figure 7| Local densities of PBS subunits. a, Local density of L_R from 7002. **b-d**, Local densities of L_{RC} proteins from 7002. **e**, Local density of L_{CM} from 7002. **f-g**, Local densities of L_C proteins from 7002. **h**, Local density of L_R from 7120. **i-l**, Local densities of L_{RC} proteins from 7120. **m**, Local density of L_{CM} from 7120. **n-o**, Local densities of L_C proteins from 7120.

Also, the article would really benefit from an additional figure comparing the cyanobacterial and eukaryotic phycobilisomes, since the comparison is done several times in the manuscript. For example, in the description of the core structure “Notably, the structure of red algal PBS contains three layers of $\alpha\beta$ trimers in the basal cylinders and two layers of $\alpha\beta$ trimers in the top cylinders” and the sentence just after “Structural comparison indicates that the equivalent layers of A1/A'1 in ...”. You are not showing any structural comparison here.

We thank the reviewer for this suggestion. We added another panel in ED Fig. 2e to show the core structure in red algae PBS.

D – Results – Maybe I missed it but I wasn’t able to identify ApcA, ApcB and ApcE; where are they in the structure?

The core cylinders are organized by $\alpha\beta$ monomers which preform as four types: ApcA and ApcB (green in Figure 1a,b,e,f); ApcA and ApcF (orange); α^{LCM} (red) and ApcB;

ApcD (magenta) and ApcB. ApcE is the core-membrane linker protein and also named as L_{CM} that is colored as red in Figure 1.

E – Results – “no Pfam01383 domain of any source was observed in the cavities of the rod hexamers from the ΔLr -PBS (Extended data Fig. 6g).” You are not showing the WT version, hence I can only believe you. Please put the WT density side-by-side with the mutant one.

We have added a panel for the WT version in ED Fig. 7g.

F – In Figure 4, the “i” “ii” “iii”... are misleading with other panel names, please replace them.

We have separated Figure 4 (previous version) into two figures (Figure 5 and Figure 6) and replace “i” “ii” “iii” with b-e and g-j.

Minor comments:

-Please remove “near atomic resolution” from the abstract. This term has no meaning anymore in cryo-EM. State the real resolution (3.5 and 3.9) or just remove it.

We thank the reviewer for the suggestion. We have removed the phrase in the revision.

-Introduction – add “about” before 2.4 billion years ago

Suggestion is well taken.

-Introduction – The first sentence should be split in two.

Suggestion is well taken.

-Results – “Both PBS contain phycocyanin (PC), allophycocyanin (APC) and linker proteins but lack phycoerythrin.” Is it important to precise that it lacks phycoerythrin? What is it?

Our intension is to compare our PBS structure with the red algae PBS, which contains phycoerythrin (PE). The difference between PC and PE lies in their bilins and the light wavelength these bilins absorb. The PE can bind phycoerythrobilin (PEB), phycoviolin (PVB) and phycourobilin (PUB) where the PC only binds PCB.

We have modified the text to avoid confusion.

-Results – In the “linker protein” paragraph, could you please explain what’s the role of FNR here. An introductory sentence may be necessary.

We thank the reviewer for the suggestion. We have added a brief introduction in the text (Page 6).

-In figure 4d – in the text, a R77A mutant is mentioned but not shown.

We have added it in the Figure 5j.

-Extended data fig.9 is not cited in the text.

It has been cited in the revision.

-The movies are nice but would benefit from annotations.

We thank the reviewer for the suggestion. We have annotated the PCBs in the movies.

-How does your single particle structures compare with the *in situ* phycobilisome structure of *synechocystis*?

Please also refer to our reply to comments #5 from reviewer 2.

These are complementary works. The structure obtained by cryo tomography (Rast et al, Nat. Plants 2019) indicates that a hemidisoidal PBS complex is the basic unit of PBS array assembly *in vivo*. The *in situ* structure also explains the advantage of a general hemidisoidal shape for the PBS to form regular arrays: two adjacent hemidisoidal PBS could stack very close to each other, potentially allowing inter-PBS energy transfer. Combined with our work, we can further explain why the light energy captures by PBS could transfer so fast to the reaction centers.

REVIEWER COMMENTS

Reviewer #2 (Remarks to the Author):

Following careful rereading of the resubmitted manuscript, I am satisfied that the authors have answered all of my queries, as well as those of the other reviewers, and that the manuscript is now acceptable for publication.

Reviewer #3 (Remarks to the Author):

Revised Manuscript NCOMMS-21-15688A

The revised manuscript entitled "Structural Insight into the Mechanism of Energy Transfer in Cyanobacterial Phycobilisomes" by Zheng et al. was improved in respect to the original version and addresses all my concerns, and, in my opinion, all the concerns of the other reviewers. I particularly appreciate that they took into account my comments on the figures organization. This work represents an important contribution to the photosynthesis field and should be published.